# Cardio-Vascular Interaction Evaluated by Speckle-Tracking Echocardiography and Cardio-Ankle Vascular Index in Hypertensive Patients

**DOI:** 10.3390/ijms232214469

**Published:** 2022-11-21

**Authors:** Tsuyoshi Tabata, Shuji Sato, Ruiko Ohno, Masahiro Iwakawa, Hajime Kiyokawa, Yukihiro Morinaga, Naoaki Tanji, Toshio Kinoshita, Kazuhiro Shimizu

**Affiliations:** 1Department of Clinical Functional Physiology, Toho University Sakura Medical Center, 564-1 Shimoshizu, Sakura 285-8741, Chiba, Japan; 2Department of Internal Medicine, Toho University Sakura Medical Center, 564-1 Shimoshizu, Sakura 285-8741, Chiba, Japan

**Keywords:** arterial stiffness, cardio-ankle vascular index, hypertension, right atrial phasic function, speckle-tracking echocardiography

## Abstract

Hypertension increases arterial stiffness, leading to dysfunction and structural changes in the left atrium (LA) and left ventricle (LV). However, the effects of hypertension on the right atrium (RA) and the right ventricle are still not fully understood. The purpose of this study was to clarify whether there is an interaction not only in the left ventricular system but also in the right ventricular system in hypertensive patients with preserved LV ejection fraction. The current retrospective observational study included patients (*n* = 858) with some risk of metabolic abnormalities (hypertension, diabetes, and dyslipidemia) who had visited our hospital and undergone echocardiography between 2015 and 2018. Among them, we retrospectively studied 165 consecutive hypertensive patients with preserved LV ejection fraction who had echocardiography performed on the same day as a cardio-ankle vascular index (CAVI) in our hospital. The phasic function of both atria was evaluated by two-dimensional speckle-tracking echocardiography. CAVI was measured using Vasela 1500 (Fukuda Denshi^®^). In the univariate analysis, CAVI was significantly correlated with LA and RA conduit function (LA conduit function, r = −0.448, *p* = 0.0001; RA conduit function, r = −0.231, *p* = 0.003). A multivariate regression analysis revealed that LA and RA conduit function was independently associated with CAVI (LA, t = −5.418, *p* = 0.0001; RA, t = −2.113, *p* = 0.036). CAVI showed a possibility that the association between heart and vessels are contained from not only LA phasic function but also RA phasic function in hypertensive patients.

## 1. Introduction

Hypertension is one of the risk factors for cardiovascular disease and can also be a cause of atherosclerosis [1]. Inadequately controlled hypertension leads to left ventricular hypertrophy and eventually to heart failure [2]. Therefore, it is important to routinely assess the actual damage of hypertension. Systemic circulation consists of the heart and vessels. While there are various methods that can analyze cardiac function, there is no good evaluation system that can analyze vascular function due to the complicated effects of blood pressure at the measurement time. The cardio-ankle vascular index (CAVI) is a non-invasive and simple method which can assess arterial stiffness independent of blood pressure at the time of measurement [3,4]. The artery dilates to the caliber to receive the blood delivered by the left ventricle during the systolic phase, and during the diastolic phase, the artery contracts to send the blood to the periphery. This function has been referred to as the “Windkessel function” or “diastolic pump”. This arterial elasticity plays an important role in systemic circulation. Therefore, decreased arterial stiffness is directly related to improvement in cardiac burden. Miyoshi reported that CAVI could reliably predict cardiovascular events in a large cohort prospective study; in patients with cardiovascular disease risk factors, those with higher CAVI had an elevated risk of cardiovascular events in the five-year observation [5]. As for the interaction between the heart and blood vessels, detailed observational studies are needed from the viewpoint of cardiovascular remodeling and cardiovascular disease development.

Recent technological advancements, such as two-dimensional speckle-tracking echocardiography (2DSTE), have enabled direct measurements of myocardial deformation indices. This technology, which can be used to detect the early stage of atrial and ventricular dysfunction before remodeling, was applied to elucidate the interaction of the left atrium (LA) and left ventricle (LV) caused by hypertension [6,7]. We have previously reported that CAVI is associated with LA phasic function [8]. However, the interaction between the right atrium (RA), right ventricle (RV), LA, LV, and the arterial tree has not yet been fully understood. The purpose of this study was to clarify whether there is an interaction not only in the left ventricular system but also in the right ventricular system in hypertensive patients with preserved LV ejection fraction (LVEF).

## 2. Results 

### 2.1. Characteristics of the Study Population

The clinical characteristics of the study population are summarized in Table 1.

The mean age was 66.5 ± 11.7 years, and 72% were men. The mean CAVI was 8.9 ± 1.2. In our study, 80 participants (48.5%) were in group B. The patients in group B were of older age, had a higher percentage of diabetes, and had higher levels of serum BNP compared to those of group A. Table 2 shows the traditional echocardiographic parameters. Patients with abnormal CAVI (group B) had a lower E/A ratio, e’, and TAPSE, but LA and RA volume index did not differ between the two groups. When applied to the LV geometry classification of the American Society of Echocardiography [9], the study subjects consisted of normal geometry (*n* = 29; group A = 19, B = 10, *p* = 0.204), concentric remodeling (*n* = 119; group A = 59, B = 60, *p* = 0.199), and concentric hypertrophy (*n* = 17; group A = 7, B = 10, *p* = 0.084). We performed the post hoc power analysis using G*Power software between group A and B. The detection power was 89% (*p* < 0.05, two tails) in this study.

### 2.2. Comparison between CAVI and 2DSTE 

As shown in Figure 1, LA reservoir strain was significantly lower in group B than in group A (20.3 ± 6.1 vs. 24.7 ± 8.0, *p* = 0.0001; Figure 1A). LA conduit strain was significantly lower in group B than in group A (8.7 ± 3.3 vs. 12.8 ± 6.1, *p* = 0.0001; Figure 1B). Similarly, RA reservoir strain was significantly lower in group B than in group A (26.7 ± 10.1 vs. 30.1 ± 11.4, *p* = 0.046; Figure 1E). RA conduit strain was significantly lower in group B than in group A (13.0 ± 5.6 vs. 15.8 ± 7.0, *p* = 0.005; Figure 1F). No significant differences between group B and group A were identified in biatrial pump strain and biventricular strain (Figure 1C,D,G,H). 

To clarify the correlation between CAVI and 2DSTE, simple regression analyses were performed, with CAVI as a dependent variable (Figure 2). CAVI was correlated with LA reservoir strain (r = −0.387, *p* = 0.0001; Figure 2A), LA conduit strain (r = −0.448, *p* = 0.0001; Figure 2B), and RA conduit strain (r = −0.231, *p* = 0.003; Figure 2F).

The RVGLS and RVFWS values were similar as follows: The RVGLS for group A was −18.9 ± 5.3 (%) and the RVFWS for group A was −18.7 ± 5.8 (%). The RVGLS for group B was −17.8 ± 4.7 (%) and the RVFWS for group A was −17.6 ± 5.3 (%). The correlation between the RVGLS and CAVI was r = 0.203, *p* = 0.009, and the RVFWS and CAVI was r = 0.142, *p* = 0.068. 

Multiple regression analyses were performed for parameters associated with LA and RA conduit function to identify independent variables. In the multiple regression analysis of LA conduit function, CAVI (standardized coefficient β = −0.375, *p* = 0.0001), SBP (standardized coefficient β = −0193, *p* = 0.006), and LVGLS (standardized coefficient β = −0.191, *p* = 0.006) were all identified as independent variables (Table 3). In contrast, the multiple regression analysis of RA conduit function, RVFWS (standardized coefficient β = 0.318, *p* = 0.0001), CAVI (standardized coefficient β = −0156, *p* = 0.036), and E/e’ (standardized coefficient β = −0.146, *p* = 0.045) were identified as independent variables (Table 4). 

## 3. Discussion

Arterial stiffness plays a key role in the pathophysiology of cardiovascular disease and is an independent predictor of cardiovascular morbidity and mortality [10,11]. The degree and duration of hypertension are known to be very important factors in the worsening of arterial stiffness as an afterload of the heart. This chronic burden subsequently leads to left ventricular hypertrophy, atrial fibrillation, and heart failure. Arterial stiffness can be evaluated by measuring the pulse wave velocity (PWV). However, the challenge with the clinical use of PWV has a limitation because PWV itself is closely dependent on blood pressure during measurement [12]. CAVI overcomes this problem and becomes the better index to assess cardiovascular interaction. To avoid the occurrence of cardiovascular disease, it is important to detect cardiovascular remodeling in the early stages. Recently, several studies on the cardiovascular interaction evaluated by CAVI have been reported [13,14,15,16]. Moreover, there have been several reports showing high CAVI in pulmonary hypertension [17,18,19,20]. Sato et al. reported that CAVI decreases with chronic thromboembolic pulmonary hypertension treatment [21]. Moreover, there are several reports on effective ways to improve CAVI, such as by administering olmesartan [22] and eplerenone [23] for hypertension; pitavastatin [24] for dyslipidemia; continuous positive airway pressure [25,26] for sleep apnea syndrome; glimepiride [27] and pioglitazone [28] for diabetes, body weight reduction [26,29], and smoking cessation [30]; and nicorandil after coronary intervention [31].

We investigated the cardiovascular interaction focusing on hypertensive patients before cardiac remodeling. This is the first report to suggest that CAVI is strongly associated with not only LA phasic function but also RA phasic function before cardiac remodeling in hypertensive patients using 2DSTE and CAVI (Figure 1 and Figure 2). Multiple regression analyses revealed that LA conduit function reflects CAVI, SBP, and LVGLS (Table 3), and that RA conduit function is associated with RVFWS, CAVI, and E/e’ (Table 4). We speculate that increased arterial stiffness, as determined by CAVI, was independently related with LA phasic function and RA phasic function in hypertensive patients with preserved LVEF.

Our current understanding of this process includes the fact that not only LA but also RA phasic dysfunction precedes cardiac structural remodeling. The results of this study indicate that increased arterial stiffness is closely associated with impaired RA conduit function. A decrease of RA conduit function has been shown for patients with increased pulmonary artery pressure [32] and LV dysfunction related to LV hypertrophy [33] and heart failure with preserved LVEF [34]. On the other hand, RA conduit function has been reported to be affected by biventricular dysfunction [35]. However, our results showed significant association between RA conduit function and CAVI compared to biventricular systolic function. This suggests the value of RA phasic function beyond systolic biventricular function.

Singh et al. [36] demonstrated that LA strain progressively worsens with the severity of diastolic dysfunction, and it could indicate an early stage of diastolic dysfunction compared to conventional echocardiographic parameters. Similar reasoning may be applied to explain the decrease in RA strain, although our knowledge of diastolic RV dysfunction is limited. In this study, we found that RVFWS and LV diastolic function (E/e’) was independently associated with RA conduit function. RA conduit dysfunction reflects the part of RV and LV dysfunction. We were able to propose the concept of cardiovascular interaction involving the right heart system in hypertensive patients using CAVI. Through this concept, we might be able to properly control the risk of future heart failure caused by hypertension.

CAVI values have been reported to be increasing by 0.05 per year in Japanese health examination data [37]. Recently, various therapeutic approaches to reduce CAVI values have been explored [38,39]. CAVI is a noninvasive method for vascular assessments. It is important to intervene appropriately before cardiac and vascular remodeling occurs.

As shown in Figure 3, the concept of cardiovascular interaction was associated with the left and right heart circulation systems through CAVI in this study. We can understand that by incorporating non-invasive and simple CAVI assessment into routine clinical practice, an increase in CAVI indicates the presence of a cardiovascular load and a decrease indicates its improvement in individual patients. The present findings together with previous research reports should facilitate improvement of our cardiovascular practice.

### Study Limitations

This study had several limitations. First, most of the patients used antihypertensive agents. ACE-I/ARBs and Ca blockers are known to have a strong effect on the regression of left ventricular hypertrophy. In this study, the percentage of those taking ACE-I/ARBs was 30% and Ca blockers, 41%. Second, our findings are based on limited data from Japanese patients. Third, no data from a simultaneous invasive evaluation of LA and RA function were available because our patients had preclinical, asymptomatic hypertension. Fourth, the present study was a retrospective cross-sectional single-center study. Fifth, 57% of study patients were treated for their hypertension. This was a retrospective observational study; we did not know the details of the duration of treatment. To further clarify this cardiovascular interaction, longitudinal studies are needed for advanced hypertensive patients.

## 4. Materials and Methods

### 4.1. Study Patients

The current retrospective observational study included patients (*n* = 858) with some risk of metabolic abnormalities (hypertension, diabetes, and dyslipidemia) who had visited the department of cardiology division of Toho University Sakura Medical Center (Chiba, Japan) and undergone echocardiography between 2015 and 2018. Among them, we retrospectively studied 165 consecutive hypertensive patients with preserved LVEF who had echocardiography performed on the same day as their CAVI. The exclusion criteria were as follows: LVEF <50%, acute myocardial infarction, old myocardial infarction, cardiomyopathy, open-heart surgery, non-sinus rhythm, atherosclerosis obliterans (ankle-brachial index, <0.9), pulmonary hypertensive disease, moderate or severe valvular disease, and strain analysis not available (Figure 4). 

Patients were defined as having hypertension if they had systolic blood pressure (SBP) ≥140 mmHg and/or diastolic blood pressure (DBP) ≥90 mmHg or were taking antihypertensive agents. SBP and DBP were measured simultaneously during CAVI measurements using Vasela 1500. We used SBP and DBP values determined from measurements at the right side in this study. According to the American College of Cardiology/American Heart Association, the detail of hypertension was classified as follows and the results are shown in Table 1. 

The presence of the following concomitant diseases was recorded: diabetes mellitus (defined as glycated hemoglobin ≥6.5% (NGSP-standardized value) or requiring antidiabetic treatment) and dyslipidemia (defined as low-density lipoprotein cholesterol concentration ≥140 mg/dL, high-density lipoprotein cholesterol concentration <40 mg/dL, triglyceride concentration ≥150 mg/dL, or requiring antihyperlipidemic treatment).

### 4.2. Measurement of the Cardio-Ankle Vascular Index

Patients were examined in a quiet room with a constant temperature; all the CAVI values were determined from measurements obtained using a vascular screening system (VaSera1500; Fukuda Denshi Co., Ltd., Tokyo, Japan) and a previously described method [3,9]. Briefly, in the supine position, cuffs were applied to a patient’s bilateral upper arms and ankles, and their head was held in the midline position. After the patient had been allowed to rest for 10 min, low cuff pressure (30–50 mmHg) was used to enable detection of both brachial and ankle pulse waves with a minimal effect on hemodynamics. Blood pressure was subsequently measured.

The CAVI value was determined with the following equation, derived from the Bramwell–Hill equation:CAVI value = a {(2*p*/Δ*P*) × In (*P*_s_/*P*_d_) *PWV*^2^} + b,
in which *P*_s_ and *P*_d_ are systolic and diastolic blood pressure, respectively; *PWV* is pulse-wave velocity from the origin of the aorta to the tibial artery-femoral artery junction; Δ*P* is the difference between systolic and diastolic blood pressure (i.e., *P*_s_ − *P*_d_); *p* is blood density; and a and b are constants. The CAVI value was adjusted for blood pressure based on the stiffness parameter *β*. We used CAVI values determined from measurements obtained from the right side in this study. 

The CAVI values were categorized into two groups according to the current recommendation for the CAVI optimal cut-off value for predicting cardiovascular disease: <9 for normal and ≥9 for abnormal [38,40]. We defined the two groups as group A (CAVI < 9) and group B (CAVI ≥ 9) and compared the results of the two groups.

### 4.3. Echocardiographic Examination

#### 4.3.1. Two-Dimensional Echocardiography

All echocardiographic examinations were carried out using a commercially available system (Vivid7, E9, S5, and S6; GE Healthcare, Boston, MA, USA) according to the American Society of Echocardiography and the European Association of Cardiovascular Imaging guidelines [41]. The Simpson’s method was used to calculate LA volume from measurements obtained in the apical 4-chamber and 2-chamber views, and RA volume was also calculated using the Simpson’s method from the apical 4-chamber view at end-systole. The LA and RA volume were then indexed according to body surface area. Early (E) and late (A) diastolic mitral inflow velocity and E/A ratio were determined by Doppler echocardiography. Tissue Doppler imaging of the septal mitral annulus was recorded to measure early diastolic velocity (e’), and the ratio of early trans-mitral valve flow velocity to mitral annular velocity (E/e’) was calculated. Tricuspid annular plane systolic excursion (TAPSE) was measured as the displacement of the lateral tricuspid annulus toward the apex during systole. 

#### 4.3.2. Speckle-Tracking Echocardiography

Biatrial and biventricular strain indices were quantitated offline, using the EchoPAC PC system, version 113 (General Electric Healthcare, Chicago, IL, USA). The software detects borders semi-automatically and tracks the LA and RA borders throughout the entire cardiac cycle. Cases of inaccurate endocardial detection were corrected manually. LA strain was determined as the average of values obtained for six LA segments in the apical 4-chamber and 2-chamber views, whereas RA strain was determined as the average of values obtained from apical 4-chamber views. The reference for zero strain was set at LV end-diastole (R-R triggering), in accordance with current recommendations [42]. Strain curves were used to evaluate three RA and LA phasic functions: reservoir, conduit, and pump function. RA and LA longitudinal strain reflects reservoir function (RAS_r_ and LASr), and RA and LA pump function (RAS_p_ and LAS_p_) corresponds to RA and LA strain at the onset of the P wave; RA and LA conduit function (RAS_c_ and LAS_c_) is the difference between the two (i.e., RAS_r_ − RAS_p_ and LAS_r_ − LAS_p_) (Figure 5A,B) [42,43]. RV global longitudinal strain (RVGLS) was also calculated based on the 4-chamber view; six segmental strain values from the RV free wall and ventricular septum were averaged. RV free wall longitudinal strain (RVFWS) was calculated by averaging each of the three regional peak systolic strains along the entire RV free wall (Figure 5C) [42,44]. LV global longitudinal strain (LVGLS) was calculated as the average negative peak of longitudinal strain in the 4-chamber, 2-chamber, and apical long-axis in accordance with current guidelines (Figure 5D) [42,43]. 

### 4.4. Statistical Analysis

For continuous variables, the two-tailed unpaired t-test or the Mann–Whitney U test was used; the data are presented as mean ± standard deviation or median (lower and upper limits of interquartile range), as appropriate. For categorical variables, the chi-square test or Fisher’s exact test was applied to the data, as appropriate. We analyzed the relationship between CAVI and various clinical parameters using Spearman’s correlation analysis. A multivariate analysis was performed using stepwise regression analysis with backward elimination to identify independent factors associated with CAVI and atrial phasic function. The SPSS software package (PASW Statistics 25, Chicago, IL, USA) was used for all statistical analyses. Statistical significance was set at *p* < 0.05. We performed the post hoc power analysis using G*Power 3 software (Germany) between group A and B [45].

## 5. Conclusions

CAVI showed a possibility that the association between heart and vessels are contained from not only LA phasic function but also RA phasic function in hypertensive patients. Hypertensive patients with high CAVI may require more attention to prevent cardiac remodeling.

## Figures and Tables

**Figure 1 ijms-23-14469-f001:**
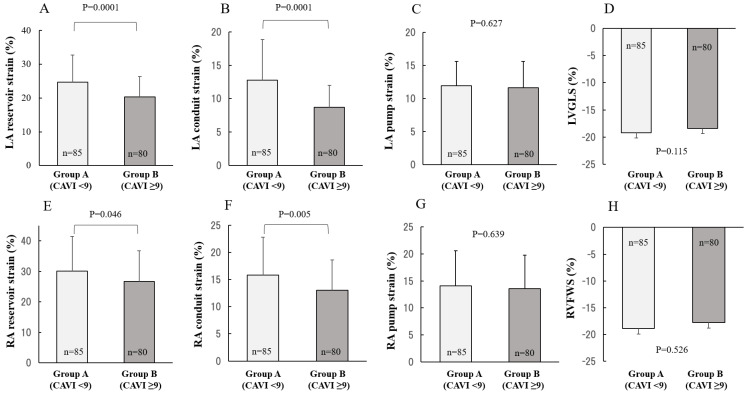
Comparison of biatrial and biventricular strain in the CAVI value. Biatrial reservoir and conduit strain in those with abnormal CAVI were significantly lower than those with normal CAVI. Data are presented as mean ± standard deviation. CAVI, cardio-ankle vascular index; LA, left atrium; RA, right atrium; LVGLS, left ventricular global longitudinal strain; RVFWS, right ventricular free wall longitudinal strain.

**Figure 2 ijms-23-14469-f002:**
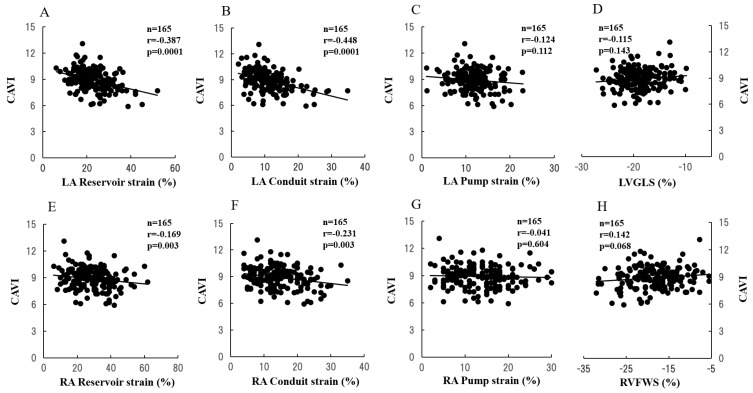
Correlations between the CAVI value and biatrial and biventricular strain. CAVI was correlated with LA reservoir strain (r = −0.387, *p* = 0.0001), LA conduit strain (r = −0.448, *p* = 0.0001), and RA conduit strain (r = −0.238, *p* = 0.003). The solid line represents the regression line. CAVI, cardio-ankle vascular index; LA, left atrium; RA, right atrium; LVGLS, left ventricular global longitudinal strain; RVFWS, right ventricular free wall longitudinal strain.

**Figure 3 ijms-23-14469-f003:**
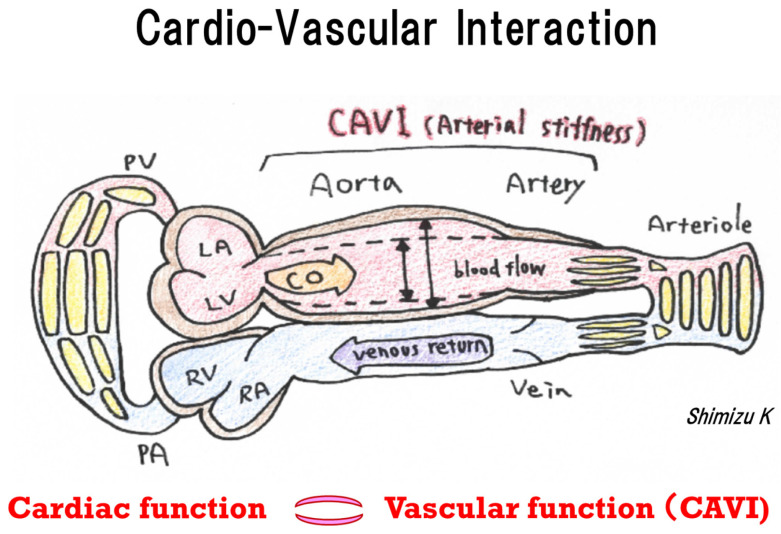
Cardiovascular interaction. Systemic circulation is pumped by the left ventricle but supported by the right ventricle. The left ventricle and all vessels are composed of the aorta, muscular artery, arteriole, and vein. Each artery has its own function. CAVI reflects vascular function of the aorta and muscle.

**Figure 4 ijms-23-14469-f004:**
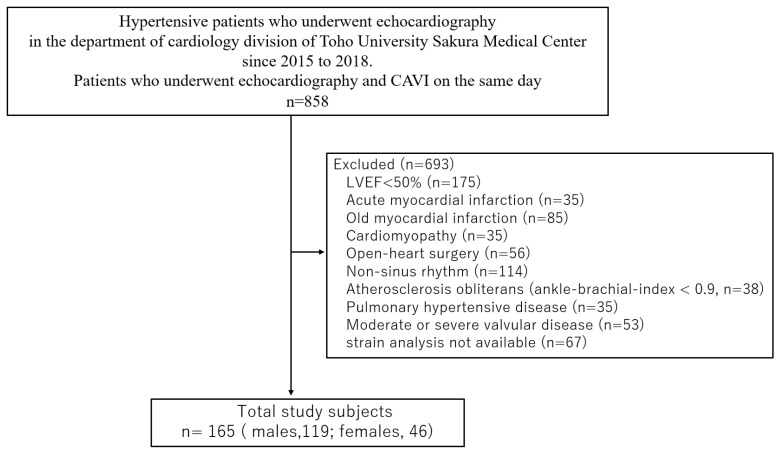
Flow chart showing patient selection. CAVI, cardio-ankle vascular index; LVEF, left ventricular ejection fraction.

**Figure 5 ijms-23-14469-f005:**
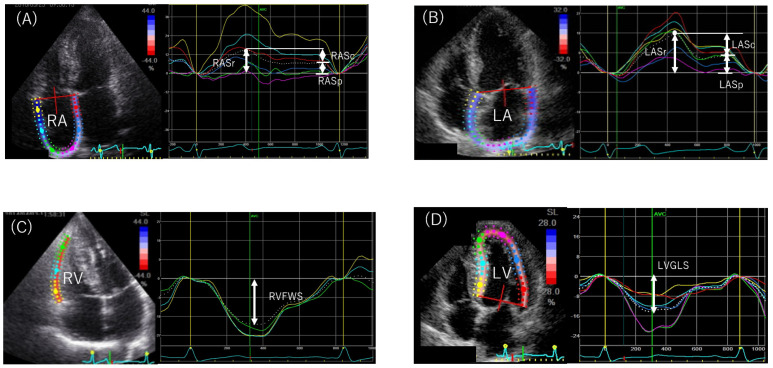
Analysis of longitudinal strain in a 4-chamber view. Dotted lines depict the average curve of the six segments. (**A**,**B**): The first positive peak of the curve is the peak atrial strain during ventricular systole, measured at the end of the reservoir phase (RASr and LASr). The peak defection is followed by a plateau and peak atrial strain in late diastole at the onset of the P wave on the electrocardiogram, just before the active atrial pump (RASp and LASp) begins. RA conduit (RASc) is calculated as the difference between RASr and RASp. LA conduit (LASc) is calculated as the difference between LASr and LASp. (**C**,**D**): Longitudinal strain of RV and LV was analyzed as the average of negative peak strain (RVFWS and LVGLS). RA, right atrium; LA, left atrium; RV, right ventricle; LV, left ventricle; RVFWS, right ventricular free wall longitudinal strain; LVGLS, left ventricular global longitudinal strain.

**Table 1 ijms-23-14469-t001:** Comparison of the clinical characteristics according to the CAVI value.

Clinical Parameter	All Subjects (N = 165)	Group A (N = 85)	Group B (N = 80)	*p* Value
age (years)	69.0 (59.5, 75.0)	64.0 (53.5, 72.5)	72.0 (66.0, 76.0)	0.0001
male, n (%)	119 (72)	61 (72)	58 (73)	0.916
BMI (kg/m^2^)	23.4 (21.2, 25.7)	23.6 (21.0, 25.9)	23.4 (21.9, 25.1)	0.872
heart rate (beats/min)	67.0 (59.0, 87.0)	68.0 (60.0, 78.0)	67.0 (58.3, 77.0)	0.393
SBP (mmHg)	135.0 (123.0, 144.0)	131.0 (119.5, 142.5)	138.0 (127.0, 144.0)	0.09
DBP (mmHg)	81.0 (74.0, 87.0)	82.0 (74.0, 88.5)	81.0 (74.5, 85.0)	0.658
BP Category				
normal, n (%)	31 (19)	21 (25)	10 (13)	0.045
elevated, n (%)	29 (18)	16 (19)	13 (16)	0.664
stage1 HT, n (%)	38 (23)	22 (26)	16 (20)	0.370
stage2 HT, n (%)	67 (41)	26 (31)	41 (51)	0.010
diabetes mellitus, n (%)	65 (39)	24 (28)	41 (51)	0.002
hyperlipidemia, n (%)	102 (62)	48 (56)	54 (68)	0.145
CAD, n (%)	34 (21)	11 (13)	23 (29)	0.012
CVD, n (%)	15 (9)	10 (12)	5 (6)	0.218
creatinine (mg/dL)	0.80 (0.65, 0.93)	0.79 (0.64, 0.90)	0.80 (0.65, 0.96)	0.462
HbA1c (%)	6.0 (5.7, 7.0)	5.9 (5.6, 6.7)	6.2 (5.8, 7.3)	0.051
BNP (pg/dL)	21.2 (11.0, 49.8)	18.0 (10.0, 31.0)	30.0 (13.9, 58.8)	0.003
CAVI	8.9 (8.2, 9.7)	8.2 (7.6, 8.7)	9.7 (9.3, 10.2)	0.0001
medications				
diuretics, n (%)	12 (7)	7 (8)	5 (6)	0.624
α-blocker, n (%)	2 (1)	1 (1)	1 (1)	0.966
β-blocker, n (%)	24 (15)	9 (11)	15 (9)	0.137
ACE/ARB, n (%)	50 (30)	23 (27)	27 (34)	0.35
calcium blocker, n (%)	67 (41)	33 (39)	34 (43)	0.631

Data are presented as mean ± standard deviation, number (%), or median (interquartile range), as appropriate. Abbreviations: BMI, body mass index; SBP, systolic blood pressure; DBP, diastolic blood pressure; BP, blood pressure; HT, hypertension; CAD, coronary artery disease; CVD, cerebrovascular disease; BNP, brain natriuretic peptide; CAVI, cardio-ankle vascular index; ACEI, angiotensin-converting enzyme inhibitor; ARB, angiotensin II receptor blocker.

**Table 2 ijms-23-14469-t002:** Comparison of the traditional echocardiographic parameters according to the CAVI value.

Traditional Echocardiographic Parameters	All Subjects (N = 165)	Group A (N = 85)	Group B (N = 80)	*p* Value
LV end-diastolic diameter (mm)	42.4 ± 5.1	42.4 ± 5.1	41.4 ± 5.1	0.188
LV end-systolic diameter (mm)	25.0 (22.0, 28.0)	25.0 (22.0, 28.0)	25.0 (22.0, 28.0)	0.898
LV ejection fraction (%)	68.0 (66.0, 70.0)	68.0 (66.0, 69.0)	67.0 (65.0, 71.0)	0.516
LV stroke volume (mL)	55.3 ± 15.8	56.8 ± 16.0	53.8 ± 15.6	0.223
LV mass index (g/m^2^)	73.0 (57.0, 91.5)	67.0 (55.0, 82.5)	75.0 (61.3, 64.8)	0.081
RWT	0.51 (0.45, 0.58)	0.50 (0.43, 0.57)	0.54 (0.46, 0.60)	0.476
E wave (cm/s)	61.0 (49.5, 72.0)	66.0 (51.0, 74.0)	57.5 (48.0, 68.0)	0.045
A wave (cm/s)	79.0 (61.5, 93.0)	73.0 (57.0, 92.0)	82.0 (69.0, 93.0)	0.116
E/A ratio	0.76 (0.61, 0.95)	0.81 (0.66, 1.07)	0.71 (0.60, 0.86)	0.003
e’ (cm/s)	5.5 (4.5, 7.0)	6.1 (4.7, 7.7)	5.2 (4.3, 6.3)	0.002
E/e’ ratio	10.6 (8.4, 14.0)	10.5 (7.9, 13.7)	11.4 (8.8, 14.3)	0.220
TAPSE (mm)	19.5 ± 4.7	20.2 ± 4.3	18.8 ± 5.0	0.047
RVFAC	44.4 ± 8.7	45.6 ± 8.5	43.1 ± 8.8	0.072
LAVI (mL/m^2^)	25.6 (20.5, 32.0)	27.0 (21.0, 32.0)	25.0 (20.0, 32.0)	0.585
RAVI (mL/m^2^)	14.0 (12.0, 18.0)	15.0 (12.5, 18.0)	14.0 (12.0, 17.8)	0.487

Data are presented as mean ± standard deviation or median (interquartile range), as appropriate. Abbreviations: RWT, relative wall thickness; E, peak early diastolic velocity of transmitral flow; A, peak atrial systolic velocity of transmitral flow; e’, peak early diastolic mitral annular motion velocity; TAPSE, tricuspid annular plane systolic excursion; RVFAC, right ventricular fractional area change; LAVI, left atrial volume index; RAVI, right atrial volume index; LV, left ventricle.

**Table 3 ijms-23-14469-t003:** Multiple regression documenting the association of LA conduit strain with clinical variables.

Independent Variable	Standard Coefficient	t Value	*p* Value
CAVI	−0.375	−5.418	0.0001
SBP	−0.193	−2.791	0.006
LVGLS	−0.191	−2.772	0.006

Variables that were not independent included: age, male sex (0:(−), 1:(+)), diabetes mellitus (0:(−), 1:(+)), hyperlipidemia (0:(−), 1:(+)), E/e’, RVFWS, LAVI, and RAVI. Abbreviations: SBP, systolic blood pressure; peptide; CAVI, cardio-ankle vascular index; E, peak early diastolic velocity of transmitral flow; e’, peak early diastolic mitral annular motion velocity; LVGLS, left ventricular global longitudinal strain; RVFWS, right ventricular free wall longitudinal strain; LAVI, left atrial volume index; RAVI, right atrial volume index.

**Table 4 ijms-23-14469-t004:** Multiple regression documenting the association of RA conduit strain with clinical variables.

Independent Variable	Standard Coefficient	t Value	*p* Value
RVFWS	0.318	4.324	0.0001
CAVI	−0.156	−2.113	0.036
E/e’	−0.146	−2.018	0.045

Variables that were not independent included, age, male sex (0:(−), 1:(+)), SBP, diabetes mellitus (0:(−), 1:(+)), hyperlipidemia (0:(−), 1:(+)), LVGLS, LAVI, and RAVI. Abbreviations: SBP, systolic blood pressure; peptide; CAVI, cardio-ankle vascular index; E, peak early diastolic velocity of transmitral flow; e’, peak early diastolic mitral annular motion velocity; LVGLS, left ventricular global longitudinal strain; RVFWS, right ventricular free wall longitudinal strain; LAVI, left atrial volume index; RAVI, right atrial volume index.

## Data Availability

The original contributions presented in the study are included in the article, and further inquiries can be directed to the corresponding author.

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
