# Peer review of "Cardio-Vascular Interaction Evaluated by Speckle-Tracking Echocardiography and Cardio-Ankle Vascular Index in Hypertensive Patients"

_ijms, 2022, doi:10.3390/ijms232214469_

Round 1

Reviewer 1 Report

The authors seek to understand the effects of hypertension on the right atrium (RA) and the right ventricle patients with preserved LV ejection fraction.

They conducted a retrospective observational study using CAVI and 2-dimensional speckle-tracking echocardiography and found CAVI to be significantly correlated with LA and RA conduit function.

However, there are serious issues with the manuscript that have to be addressed.

Major points:

·         The purpose of the study to understand the correlation with the RA and RV and its scientific relevance is not entirely clear.

·         A proper statistical analysis of the sample size has not been performed.

·         The study subjects and the entire methods section has been based on first authors’ previous publication (https://www.ncbi.nlm.nih.gov/pmc/articles/PMC8757747/), where they already investigated the correlation with LA and LV function. There is a lot of overlap from this publication.

·         While these studies utilize the same study population, the values and P values in the echocardiographic parameters mentioned in Table 2 is different from the previous publication. This discrepancy is highly concerning.

·         The discussion section lacks a detailed explanation of the obtained results and their significance along with relevant references.

Author Response

Response letter

Dear Reviewer, thank you very much for your efforts on my manuscript.

 Tsuyoshi Tabata and Kazuhiro Shimizu.

Reviewer 1

Thank you very much for your appropriate comments.

   The purpose of the study to understand the correlation with the RA and RV and its scientific relevance is not entirely clear.

Thank you.

The concept that hypertension exerts its influence on the whole circulation is the topic in this study. As a method for quantitative assessment of arterial stiffness, pulse wave velocity (PWV) has been used for the last several decades and it was thought to be a kind of surrogate marker of arteriosclerosis. PWV depends inherently on blood pressure (BP) at the time of measurement. In 2006, the cardio-ankle vascular index (CAVI) was developed as an arterial stiffness index, which was derived from the stiffness parameter beta theory with the application of the Bramwell–Hill equation. According to the previous studies about CAVI using α1- blocker and β1-blocker, it is essentially independent from BP at the time of measurement. Therefore, CAVI is the best evaluation system for blood pressure assessment at the present time.

Previously, we reported the association between CAVI and left atrium. Thereafter, we thought the cardio-vascular interaction might be related not only to the left heart circulation system but also to the right heart system in hypertensive patients. We were able to articulate the concept of cardio-vascular interaction involved right heart system about hypertensive patients.

   A proper statistical analysis of the sample size has not been performed.

Thank you.

We performed the post hoc power analysis using G*Power 3 software (Germany) between group A and B. We added these sentences in Statistical analysis part (line:178-179). We performed the post hoc power analysis using G*Power software between group A and B. The detection power was 89% (p<0.05, two tails) in this study.   We added these sentences in RESULTS part (line:193-195).

   The study subjects and the entire methods section has been based on first authors’ previous publication (https://www.ncbi.nlm.nih.gov/pmc/articles/PMC8757747/), where they already investigated the correlation with LA and LV function. There is a lot of overlap from this publication.

Thank you.

The previous our study was conducted only in LA and LV function. In this time, we reanalyzed whether the increased arterial stiffness extends to the right heart system. These facts were stated on p. 4, lines 73-74.

      While these studies utilize the same study population, the values and P values in the echocardiographic parameters mentioned in Table 2 is different from the previous publication. This discrepancy is highly concerning.

Thank you.

In our previous study, we stratified patients into 3 groups according to Japanese standard deviation of CAVI value for age. In this time, we stratified 2 groups according to the current recommendation cut-off value for atherosclerosis: (<9 for normal, and ≥9 for advanced atherosclerosis). Therefore, there are some parts which differ from the previous analysis.

The discussion section lacks a detailed explanation of the obtained results and their significance along with relevant references.

Thank you.

CAVI values have been reported to increase by 0.05 per year in Japanese health examination data [44]. Recently, various therapeutic approaches to reduce CAVI values have been explored [11, 45]. CAVI is the noninvasive method for vascular assessments. It is important to intervene appropriately before cardiac and vascular remodeling. CAVI is the noninvasive method for vascular assessments.

We added these sentences in discussion part and renumbered the literature (line:267-271).

Reference 11: Saiki A, Ohira M, Yamaguchi T, Nagayama D, Shimizu N, Shirai K, et al. New Horizons of Arterial Stiffness Developed Using Cardio-Ankle Vascular Index (CAVI). J Atheroscler Thromb. 2020; 27: 732-748. doi: 10.5551/jat.RV17043.

Reference 44: Namekata T, Suzuki K, Ishizuka N, Shirai K: Establishing baseline criteria of cardio-ankle vascular index as a new indicator of arteriosclerosis: a cross-sectional study. BMC Cardiovasc Disord, 2011; 11: 51. doi: 10.1186/1471-2261-11-51.

Reference 45: Wright LM, Dwyer N, Wahi S, Marwick TH. Association with right atrial strain with right atrial pressure: an invasive validation study. Int J Cardiovasc Imaging, 2018 Oct;34(10):1541-1548. doi: 10.1007/s10554-018-1368-3.

Reviewer 2 Report

Overall well done.

Some minor issuses

- the sentence "We defined the two groups as Group A (CAVI <9) and Group B (CAVI
9) and compared the results of the two groups
" in section 3.1 belongs to "Materials & Methods"

- The regression lines in most of the figures are almost horizontal. This indicates that minor variations in CAVI could result in large variations in parameters such as RVFWS. However, the dots are arranged as a cloud. How useful can these correlations be for clinicians?

- reference 2: 'Cambell' must be 'Campbell'

- Fig 5 should be redrawn by a professional.

Author Response

Response letter

Dear Reviewer, thank you very much for your efforts on my manuscript.

 Tsuyoshi Tabata and Kazuhiro Shimizu.

Reviewer 2

Thank you very much for your appropriate comments.

the sentence "We defined the two groups as Group A (CAVI <9) and Group B (CAVI ≥9) and compared the results of the two groups" in section 3.1 belongs to "Materials & Methods"

Thank you.

CAVI values were categorized into two groups according to the current recommendation for CAVI optimal cut-off value for predicting cardiovascular disease: (<9 for normal, and ≥9 for abnormal) [10,11]. We defined the two groups as Group A (CAVI <9) and Group B (CAVI ≥9) and compared the results of the two groups.

We moved these sentences in Materials & Methods part (line:124-127).

The regression lines in most of the figures are almost horizontal. This indicates that minor variations in CAVI could result in large variations in parameters such as RVFWS. However, the dots are arranged as a cloud. How useful can these correlations be for clinicians?

Thank you.

CAVI values have been reported to increase by 0.05 per year in Japanese health examination data [44]. Recently, various therapeutic approaches to reduce CAVI values have been explored [11, 45]. CAVI is the noninvasive method for vascular assessments. It is important to intervene appropriately before cardiac and vascular remodeling. CAVI is the noninvasive method for vascular assessments.

We added these sentences in discussion part and renumbered the literature (line:267-271).

Reference 11: Saiki A, Ohira M, Yamaguchi T, Nagayama D, Shimizu N, Shirai K, et al. New Horizons of Arterial Stiffness Developed Using Cardio-Ankle Vascular Index (CAVI). J Atheroscler Thromb. 2020; 27: 732-748. doi: 10.5551/jat.RV17043.

Reference 44: Namekata T, Suzuki K, Ishizuka N, Shirai K: Establishing baseline criteria of cardio-ankle vascular index as a new indicator of arteriosclerosis: a cross-sectional study. BMC Cardiovasc Disord, 2011; 11: 51. doi: 10.1186/1471-2261-11-51.

Reference 45: Wright LM, Dwyer N, Wahi S, Marwick TH. Association with right atrial strain with right atrial pressure: an invasive validation study. Int J Cardiovasc Imaging, 2018 Oct;34(10):1541-1548. doi: 10.1007/s10554-018-1368-3.

reference 2: 'Cambell' must be 'Campbell'

Thank you.

reference 2: We corrected "Cambell" to "Campbell".

Fig 5 should be redrawn by a professional.

   Thank you.

   The editor has asked for a response within 10 days. We will consider drawing.

Round 2

Reviewer 1 Report

Thank you for clarifying and addressing my concerns. There is still only one minor point to be addressed.

·         CAVI values have been reported to increase by 0.05 per year in Japanese health examination data [44]. Recently, various therapeutic approaches to reduce CAVI values have been explored [11, 45]. CAVI is the noninvasive method for vascular assessments. It is important to intervene appropriately before cardiac and vascular remodeling. CAVI is the noninvasive method for vascular assessments.

This addition to the discussion section does not fit properly and the sentence ‘CAVI is the noninvasive method for vascular assessments.’ has been repeated twice. Please include some lines on the clinical benefits and consequences of monitoring, particularly RA phasic and conduit function and how this manuscript improves the prevailing clinical practice. Reasoning similar to lines 272-278, but focused on RA, which is the highlight of this manuscript.  

Author Response

Response letter

Reviewer 1

Thank you very much for your efforts on our manuscript.

Tsuyoshi Tabata and Kazuhiro Shimizu.

This addition to the discussion section does not fit properly and the sentence ‘CAVI is the noninvasive method for vascular assessments.’ has been repeated twice. Please include some lines on the clinical benefits and consequences of monitoring, particularly RA phasic and conduit function and how this manuscript improves the prevailing clinical practice. Reasoning similar to lines 272-278, but focused on RA, which is the highlight of this manuscript.  

Thank you.

Following your advice, we deleted these repeated sentence (Line:274-275).

CAVI is the noninvasive method for vascular assessments. It is important to intervene appropriately before cardiac and vascular remodeling. CAVI is the noninvasive method for vascular assessments.

We added following sentences in DISCUSSION part. (Line:266-271)

In this study, we found that RVFWS and LV diastolic function (E/e’) was found independently associated with RA conduit function. RA conduit dysfunction reflects the part of RV and LV dysfunction. We were able to propose the concept of cardio-vascular interaction involved right heart system about hypertensive patients using CAVI. Imaging this concept, we might be able to control properly the risk of future heart failure caused by hypertension.
